# Analysis of differences and commonalities in wildlife hunting across the Africa-Europe South-North gradient

Mona Estrella Bachmann [1,2,3]*, Lars Kulik[1], Tsegaye Gatiso[4], Martin Reinhardt Nielsen[3], Dagmar Haase[2,5], Marco Heurich[6,7], Ana Buchadas[2,8], Lukas Bösch[9], Dustin Eirdosh[1], Andreas Freytag[10], Jonas Geldmann[11], Arash Ghoddousi[2], Thurston Cleveland Hicks[12], Isabel Ordaz-Németh[13], Siyu Qin[2], Tenekwetche Sop[1], Suzanne van Beeck Calkoen[6,14], Karsten Wesche[15,16,17], Hjalmar S. Kühl[1,16]

1 Max Planck Institute for Evolutionary Anthropology, Leipzig, Germany, 2 Geography Department, Humboldt-University Berlin, Berlin, Germany, 3 Department of Food and Resource Economics, University of Copenhagen, Copenhagen, Denmark, 4 Institute for Food and Resource Economics, Faculty of Agriculture, Bonn University, Bonn, Germany, 5 Helmholtz Centre for Environmental Research (UFZ), Leipzig, Germany, Leipzig, Germany, 6 Chair of Wildlife Ecology and Management, Faculty of Environment and Natural, Albert Ludwigs University Freiburg, Freiburg, Germany, 7 Faculty of Applied Ecology, Agricultural Sciences and Biotechnology, Institute for forest and wildlife management, Campus Evenstad, Koppang, Norway, 8 Integrated Research Institute on Transformations of Human-Environment Systems (IRI THESys), Berlin, Germany, 9 Institute for Sociology, University Leipzig, Leipzig, Germany, 10 Faculty of Economics and Business Administration, Friedrich Schiller University Jena, University of Stellenbosch; CESifo Research Network, Jena, Germany, 11 Center for Macroecology, Evolution and Climate, Globe Institute, University of Copenhagen, Copenhagen, Denmark, 12 Faculty of Artes Liberales, University of Warsaw, Warsaw, Poland, 13 Re:wild, Austin, Texas, United States of America, 14 Department of Visitor Management and National Park Monitoring, Bavarian Forest National Park, Grafenau, Germany, 15 Senckenberg Museum für Naturkunde Görlitz, Görlitz, Germany, 16 International Institute Zittau, Technische Universität Dresden, Zittau, Germany, 17 German Centre for Integrative Biodiversity Research (iDiv) Halle-Jena-Leipzig, Leipzig, Germany

* mona.bachmann@ifro.ku.dk

**Data Availability Statement:** The data and scripts underlying the analyses and plots are available in the Appendix.

## Abstract

Hunting and its impacts on wildlife are typically studied regionally, with a particular focus on the Global South. Hunting can, however, also undermine rewilding efforts or threaten wildlife in the Global North. Little is known about how hunting manifests under varying socioeconomic and ecological contexts across the Global South and North. Herein, we examined differences and commonalities in hunting characteristics across an exemplary Global South-North gradient approximated by the Human Development Index (HDI) using face-to-face interviews with 114 protected area (PA) managers in 25 African and European countries. Generally, we observed that hunting ranges from the illegal, economically motivated, and unsustainable hunting of herbivores in the South to the legal, socially and ecologically motivated hunting of ungulates within parks and the illegal hunting of mainly predators outside parks in the North. Commonalities across this Africa-Europe South-North gradient included increased conflict-related killings in human-dominated landscapes and decreased illegal hunting with beneficial community conditions, such as mutual trust resulting from community involvement in PA management. Nevertheless, local conditions cannot outweigh the strong effect of the HDI on unsustainable hunting. Our findings highlight regional challenges that

**Funding:** This project was hosted by the Max Planck Institute for Evolutionary Anthropology and funded by the German Center for Integrative Biodiversity Research (iDiv; DFG FZT 118) (H. K, T. G) and the Robert Bosch Foundation (grant number 32.5.8043.0016.0) (H. K). The funders had no role in study design, data collection and analysis, decision to publish, or preparation of the manuscript.

**Competing interests:** The authors have declared that no competing interests exist.

**Abbreviations:** AIC, Akaike information criterion; BHRM, Bayesian hierarchical regression model; HDI, Human Development Index; HWC, human–wildlife conflict; IUCN, International Union for Conservation of Nature; LPD, Living Planet Database; MCMC, Markov chain Monte Carlo; NGO, nongovernmental organization; PA, protected area; PAME, Protected Area Management Effectiveness.

require collaborative, integrative efforts in wildlife conservation across actors, while identified commonalities may outline universal mechanisms for achieving this goal.

## Introduction

Biodiversity loss and the rise of zoonotic diseases have reinforced global calls to mitigate the unsustainable exploitation of wildlife [1]. Scientific activity and international scrutiny commonly focus on the Global South [1], where poverty-driven and excessive hunting leads to dramatic declines in mammal and bird populations [2,3]. In the more affluent nations of the Global North, growing and returning populations of large herbivores and predators such as grey wolves (*Canis lupus*) and European bison (*Bison bonasus*) are symbols of conservation success [4]. Nevertheless, both illegal and intensive legal hunting limit population recovery and have even eradicated populations [4–7]. For many protected predators, illegal persecution remains a major cause of mortality and is even socially accepted in some parts of Europe [4,5].

Despite these similarities, most research remains geographically "segmented" in a firmly anchored South-North divide [2,8], and little is known about how hunting unfolds across the Global South and North under varying socioeconomic contexts. The function of hunting can extend far beyond the traditional provisioning of meat. Hunting underlies economies and cultures, serves as recreation, and aims at biodiversity conservation or the eradication of invasive or conflict-prone species [9]. These human–wildlife relationships are regionally manifesting phenomena shaped by a complex set of needs, social and cultural constructs, moral values, material realities, and political and historical characteristics that vary over time, locations, and human communities [10]. Thus, conservation measures must build on an understanding of the local context while also integrating perspectives gleaned from large-scale assessments that explores contextual gradients that might provide general insights [11]. Additionally, hunting has played an essential role in human evolution and is practised in most modern societies, from hunter-gatherers to rural and urban dwellers and across all social classes [9,10]. Based on this extended evolutionary perspective, a large-scale, comparative analysis of hunting patterns across socioeconomic contexts will help us to better understand the mechanisms that outline generalisable pathways and solutions within the complex phenomenon of wildlife overexploitation.

The Global South and North terminology describes a specific socioeconomic gradient that has been shaped by the recent historical division into largely colonised and colonial states. Even today, one-fourth of the world's population in the Global North controls four-fifths of global income [12]. The state of biodiversity appears in different stages, as more wildlife species have been exterminated in the Global North [4,13]. Nevertheless, the conservation and science sectors are largely dominated by Global North actors, concepts, and worldviews [14,15]. Both double standards and long-standing inequalities lead to accusations of cultural imperialism and neocolonialism in the Global South [14,16,17]. The firm South-North divide in research and discourse can foster ethnocentric representations of the Global South as the "other" of the norm, which can legitimise controversial conservation approaches to hunting that would be unacceptable in the Global North. Such approaches include "fortress protection", "green militarisation", and "shoot-on-sight policies" [14,18,19]. Implicit biases about poverty, inequality, historical grievances, and colonial and racist discourses may still shape perceptions of hunting and poaching [18]. Conversely, comparisons made across this contrasting context can further our understanding of the differences, commonalities, and universal mechanisms within the phenomenon of hunting [20]. Such a comprehensive perspective on the processes around

hunting across the contrasting economic and political contexts and path dependencies of the Global South and North allows us to outline local and universal processes and corresponding problems and possible solutions. Ultimately, understanding the local versus generalisable dimension of hunting fosters cross-contextual learning and allows an equitable conservation debate between actors in the Global North and South.

Herein, we aim to understand the differences and commonalities of wildlife hunting across a contextual and exemplary Global South-North gradient using Africa and Europe as examples. Given the steep socioeconomic gradient between Africa and Europe, its relationship with threats to wildlife and the multifunctionality of hunting, we expect fundamental differences in all 4 basic characteristics of hunting: (1) why, (2) what, and (3) where people hunt, as well as (4) how unsustainable hunting can be mitigated. The question remains whether commonalities exist despite the contrasting context. We focus on protected areas (PAs) and their immediate surroundings as an interface of biodiversity protection and resource use. PAs are an important cornerstone for biodiversity conservation [2]. Nevertheless, illegal exploitation within parks and human–wildlife conflict (HWC) beyond PA boundaries pose a major threat to biodiversity [2,21]. Hunting is frequently permitted for wildlife management, and it is usually subject to regulations designed to guarantee sustainability. However, legal hunting can also alter wildlife communities and movements, shape life histories by artificial selection, and decimate populations to a socially acceptable but ecologically fragile minimum [6,22,23]. Herein, we aim to conduct a holistic assessment that extends beyond the simplistic binary view of "legal and sustainable" versus "illegal and unsustainable" to understand the diverse manifestations of hunting and its varying impacts on wildlife. Therefore, we define hunting as the entirety of activities involved in the management and pursuit of wildlife [9]. Unsustainable hunting encompasses activities identified by PA managers as not being in harmony with PAs, such as removals that exceed population growth and disruptive hunting practices.

Current global PA assessments usually capture hunting as a homogeneous threat [24]. To obtain a more detailed assessment, particularly with regard to varying socioeconomic contexts, we conducted face-to-face interviews with 114 PA managers in 25 African and European countries (Fig 1A and Table A in S1 Appendix). We used the Human Development Index (HDI) to approximate the different socioeconomic contexts and path dependencies across the 2 continents (Fig 1), which we refer to as Africa-Europe South-North gradient (S-N gradient) (Fig 1D). We selected PAs to cover a broad HDI range, from a country with one of the lowest HDI values worldwide, namely, the Central African Republic, to a country with one of the highest values, namely, Germany. To understand (1) why, we examined (a) the predominant function of hunting, using "economic" (subsistence, commercial hunting), "sociocultural" (nonmarket cultural, social, recreational aspects), and "ecological" (population management, killing due to HWC) motivations (analysis i, Table 1) [9]. We hypothesised that a change in the motivation for hunting across the studied PAs would reflect changing socioeconomic conditions (all hypotheses in Table B in S1 Appendix). We expected a strong economic function of hunting in areas with low HDI values due to a reliance on hunting for livelihoods [3] that is gradually replaced by other functions with an increasing HDI. We also assessed the impact of hunting as a human–wildlife interaction and, more specifically, in response to HWC (analysis ii). We expected a decreasing threat of hunting along the S-N gradient since increasing prosperity reduces hunting pressure related to maintaining livelihoods [3]. In contrast, we expected the effects of HWC to be higher in the Global South due to the higher proportionate costs to households from HWC [17]. To answer (2) what, we compiled information about species targeted by legal and illegal hunting (including HWCs) and examined whether ecological parameters such as trophic level influence the likelihood of being threatened by illegal hunting (analysis iii). We expected changes in the species targeted by hunting to be relative to changing

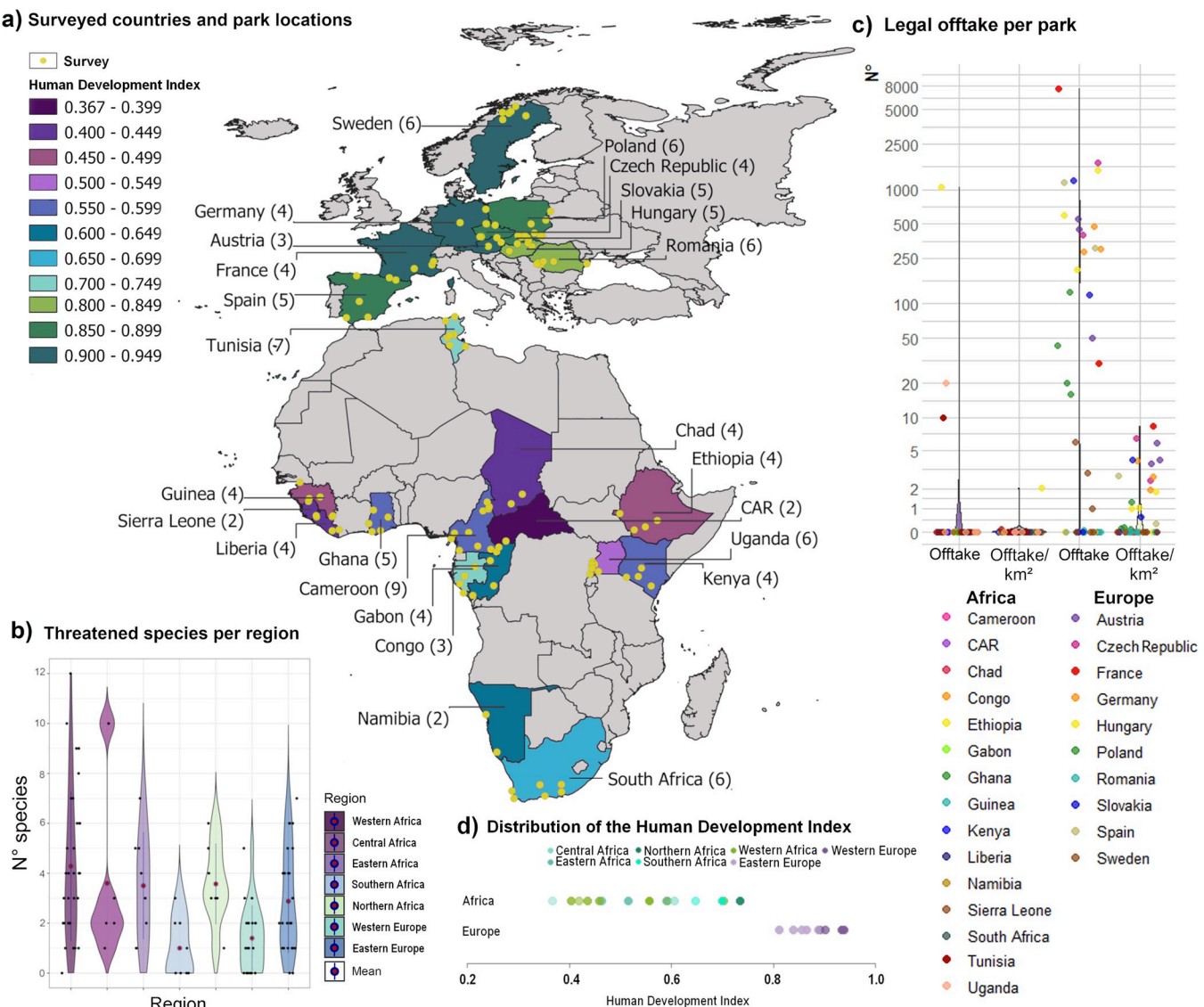

**Fig 1. Overview of sample locations, legal setting, and threatened species.** a) Map of sampled PAs (yellow dots) and HDI per surveyed country. The Global South-North divide in our data is >0.75, defined as the so-called Global North (Europe, green), and <0.75, defined as the Global South (Africa, purple–blue) (the data underlying this figure can be found in S1 Data) (source: https://public.opendatasoft.com/explore/dataset/world-administrative-boundaries/export/). (b) Violin plots displaying the absolute number of species threatened by hunting listed by PAs across regions, black dots = number of listed threatened species per park, blue and red dots = mean per region, blue line = interquartile range, each side of the blue line is a kernel density estimation of the data (S2 Data). (c) Legal offtake per PA in absolute numbers (left) and numbers per km² per continent (right) (Table U in S1 Appendix and S3 Data). (d) Distribution of the HDI along the surveyed regions. The distribution shows the socioeconomic gradient covered by our survey along the Global South-North, or respectively, the Africa-Europe South-North gradient (S-N gradient) (S1 Data). HDI, Human Development Index; PA, protected area.

functions of hunting across socioeconomic contexts, i.e., more consumable species hunted under lower HDI values [9]. To understand (3) where, we examined illegal hunting and whether this threat to wildlife tends to be located more within or outside PA boundaries (analysis iv). We expected a decreasing threat from illegal hunting within parks with increasing socioeconomic conditions since an increase in prosperity might decrease economic pressure and thus lower the willingness to risk hunting illegally within parks. To investigate (4) how unsustainable hunting can be mitigated, we used indices that reflect local community

**Table 1. Overview of the 7 response variables and predictors in each of the 9 models.**

| Analyses | Response | Predictors |
|---|---|---|
| i | Function of hunting: Hunting index across all functions (Hunting index: high (major motivation), low (neglectable motivation) | HDI $^x$ functions of hunting (ecological, economic, social), human population density $^x$ functions of hunting, community characteristics $^x$ functions of hunting, CONTROL: PA size, country (S1 Script) |
| ii | Threat rating "hunting/poaching", (legal, illegal). (Threat: high, low) | HDI, human population density, community characteristics, protection-based interventions, community-based interventions, CONTROL: PA size, country, abundance mammals/birds (S2 Script) |
| ii | Threat rating "killing because of human–wildlife conflict", (legal, illegal). (Threat: high, low) | HDI, human population density, community characteristics, protection-based interventions, community-based interventions, CONTROL: PA size, country, abundance mammals/birds (S2 Script) |
| iii | Threat of illegal hunting by trophic level of threatened species. Differentiating between predators, including raptors, predatory mammals, versus nonpredatory, containing primates, apes, omnivorous, frugivorous, insectivorous mammals and birds. (Threatened: yes, no) | HDI, human population density, community characteristics, protection-based interventions, community-based interventions, CONTROL: PA size, country, abundance predators (S3 Script) |
| iv | Rating of the threat to wildlife from parks through illegal hunting (killing/poaching/poisoning) within the administrative PA boundaries. (Threat: high, low). | HDI, human population density, community characteristics, protection-based interventions, community-based interventions, CONTROL (PA size, country, abundance mammals/birds (S4 Script) |
| iv | Rating of the threat to wildlife from parks through illegal hunting (killing/poaching/poisoning) outside the administrative PA boundaries. (Threat: high, low). | HDI, human population density, community characteristics, protection-based interventions, community-based interventions, CONTROL: PA size, country, abundance mammals/birds (S4 Script) |
| iv | Rating of the threat to wildlife from parks through hunting (killing/poaching/poisoning) within the administrative PA boundaries, (Threat: high, low). | Replacement of community attributes by its index components: "trust", "attitudes", "culture", HDI, human population density, protection-based interventions, community-based interventions, CONTROL: PA size, country, abundance mammals/birds (S4 Script) |
| iv | Rating of the threat to wildlife from parks through hunting (killing/poaching/poisoning) outside the administrative PA boundaries. (Threat: high, low). | Replacement of community attributes by its components: "trust", "attitudes", "culture", HDI, human population density, protection-based interventions, community-based interventions, CONTROL: PA size, country, abundance mammals/birds (S4 Script) |
| v | Community characteristics (beneficial characteristics: high, low) | Index components of community-based interventions: provision of economic benefits to the community, implementation of livelihood projects, scale of local inclusion, implementation of environmental awareness programs, CONTROL: population density, country, continent (S5 Script) |

characteristics (nature-friendly cultures, positive attitudes towards wildlife and PAs, positive relationships with PA management) and implemented conservation interventions (protection- and community-based) as predictors (see details of construction of indices in Table B in S1 Appendix). We expected a positive effect of supportive local community attributes, as such attributes can enhance conservation outcomes [25]. We further included regional human population density, which is known to affect human–wildlife relationships [26]. Given this juxtaposition of differences and commonalities, we aimed to identify regional challenges and

general mechanisms that ultimately can help to develop effective wildlife conservation measures across the studied Africa-Europe South-North gradient.

## Results

We collected comprehensive information by conducting face-to-face interviews with managers of 114 PAs in 10 European ($n$ = 48) and 15 African countries ($n$ = 66) (Fig 1A and Table A in S1 Appendix). Most PAs fell into IUCN management category II (69%; Africa 65%, Europe 72%), while the remaining PAs were in IUCN categories III to VI or had no assigned category (see Table A in S1 Appendix). African PAs were larger on average (2,253 km², median = 973 km², range = 26 to 21,812 km²) than European PAs (570 km², median = 330 km², range = 13 to 3,446 km²), while European PAs were on average older (45 years, median = 36, range = 16 to 109 years) than African PAs (36 years, median = 32 years, range = 8 to 87 years).

Threats from hunting were not confined to the Global South. Of the 114 interviewed PA managers, only 8 of 48 (17%) European and 4 of 66 (8%) African parks did not list any species threatened by hunting (Fig 1B). African parks listed on average significantly more species as threatened by hunting (3.6 ± 2.8) than European parks (2.2 ± 1.9) (t = 3.001, $p$-value = 0.003). When we differentiated between subregions, we found a more heterogeneous picture, with the lowest number of species listed in Southern Africa and Western Europe (Fig 1B). Although most parks fell under IUCN management category II (Table A in S1 Appendix), legal hunting for population control was more common in Europe (25 of 48 PAs, 52%) than in Africa (3 of 66 PAs, 4.55%), with an average offtake of 389 ± 1189 animals/park (1.08 ± 1.72 animals/km²) in Europe and 17 ± 133 animals/park (0.04 ± 0.26 animals/km²) in Africa. The maximum was reached in a PA in France, with on average, 7,104 hunted wild boars (*Sus scrofa*) and 510 hunted deer per year (Fig 1C and Table U in S1 Appendix). The average offtake seemed to be lower when at least 1 large predator species was present (mean = 0.36 animals/km²) compared

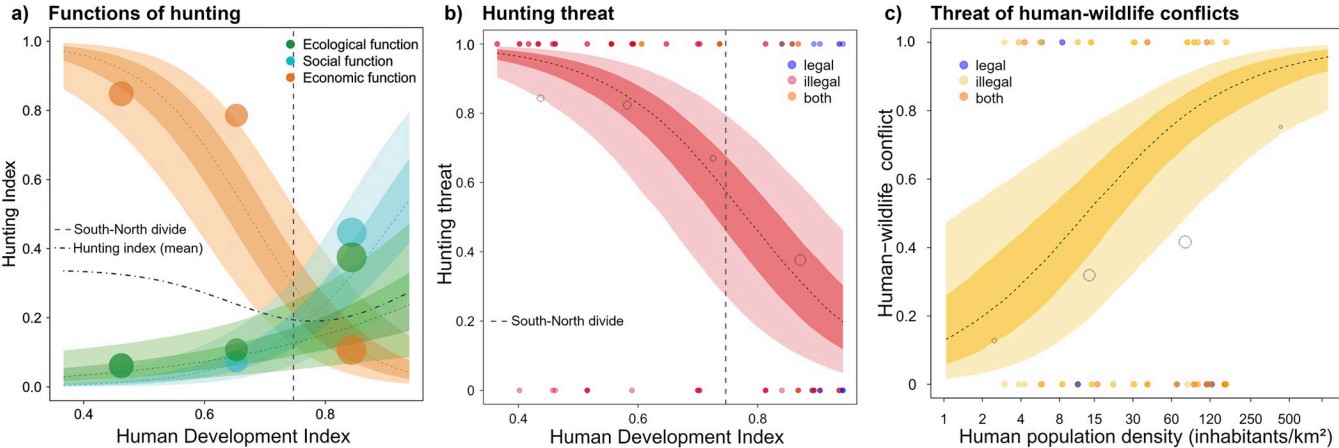

**Fig 2. Why people hunt.** (a) Functions of hunting as an interaction between HDI and the respective hunting functions, ecological (population control, HWCs), social (entertainment, sociocultural hunting), economic function (subsistence, commercial hunting). The hunting index refers to whether the function is one of the main motivations or only a negligible motivation for hunting in the PA. The categories "ecological function" and "social function" displayed a larger hunting motivation with an increasing HDI. In contrast, the economic category revealed a clearly decreasing hunting motivation; the functions split at the Global South-North divide (grey dashed line), black dotted line = mean (the data underlying this figure can be found in S4 Data). Human–wildlife interactions. (b) Hunting: The probability of a high threat through hunting decreased along the S-N gradient (S5 Data). c) HWC-driven hunting: The probability of high threats due to HWCs in relation to human population density (inhabitants/km²) revealed an increasing trajectory with human densities (S6 Data). The dashed line depicts the expected mean of the predicted posterior distribution; the coloured areas depict the 33% and 66% credibility intervals. Transparent points are binned probabilities. The size of the bubbles corresponds to the respective number of PAs. Filled points: original data; vertical grey dashed line: Global South-North divide of our data (0.75). HDI, Human Development Index; HWC, human–wildlife conflict; PA, protected area.

to in their absence (mean = 0.74 animals/km$^2$). However, these differences were not significant (t = 1.11, *p*-value = 0.272).

## Why people hunt

Regarding the question of why people hunt, our Bayesian regression models revealed variations in the prevailing function of hunting across the S-N gradient (Fig 2A). Over the studied range of the HDI, we found that our 3 hunting functions show different trajectories. The "ecological function" revealed a slightly stronger prevalence as the HDI increased. This is even more pronounced for the category "social function". The "economic function" revealed a completely different pattern, with clearly decreasing prevalence with increasing HDI (Fig 2A and Table C in S1 Appendix). The effects were nonlinear, with an observed threshold and division at HDI>0.75 (European levels) (Fig 2A, HDI>0.75, grey vertical line), corresponding to the transition from Africa to Europe. The average index across all functions remained relatively equal over the S-N gradient (Fig 2A, horizontal dashed black line).

Regarding human–wildlife interactions, 37.5% (42 of 112) of park managers reported high threat levels to wildlife by hunting due to unsustainable offtake rates. These high ratings referred 4 times (9.6%) to legal hunting, 30 times (71.4%) to illegal hunting, and 8 times (19%) to both where the effects were inseparable (Fig 2B). Increasing HDI levels were the strongest predictor, thereby reducing the probability of high threat scoring by hunting (Fig 2B and Table E in S1 Appendix, estimate = −1.58, CI = [−2.59, −0.65]). The point probability halved when reaching Europe (HDI>0.75) (Fig 2B), which suggests that unsustainable hunting also occurred in Europe, but to a lesser extent. Rankings for high hunting pressure were more evenly distributed across the S-N gradient, while absences were more frequent with higher HDI levels. Wildlife abundances were slightly negatively associated with high hunting threats (estimate = −0.76, CI = [−1.69, 0.06]). The strong effect of the HDI remained when only considering cases of illegal hunting (Table G in S1 Appendix, HDI: estimate = 1.7, CI = [−2.79, −0.62]).

Considering HWC, including all species, 20.6% (20 of 97) of the parks reported high threat levels by HWC-driven hunting, whereby only 2 of these high ratings referred to a legal setting. Human population density was the strongest predictor, thereby increasing the prevalence of HWC (Fig 2C and Table H in S1 Appendix, estimate = 0.96, CI = [−0.06, 2.04]). Wildlife abundance trends over the last 10 years, which were included as a control predictor in the analyses (Table B in S1 Appendix), were negatively associated with higher levels of HWC (estimate = −0.59, CI = [−1.57, 0.37]). We found similar effects for the 2 models (hunting, HWC) irrespective of the grouping (considering levels of "high" and "very high" or "moderate", "high", and "very high") (Table F and I in S1 Appendix).

## What people hunt

Species were listed as threatened by illegal hunting (including all lethal interactions, such as snaring, shooting, poisoning, or beating to death) 316 times, belonging to 117 different species (Fig 3A). Unfortunately, it is not possible to give a proportional figure here, as the total number of species present in a PA was often unknown. Within the guilds found on both continents, herbivores were more threatened by illegal hunting (including HWC) in Africa, and predators were more threatened in Europe (Fig 3A). Overall, wild boars and red deer (*Cervus elaphus*) were the species hunted most legally (Fig 3C). Although we unfortunately did not obtain sufficient data to compare the total biomass taken, legally or illegally hunted, in Europe and Africa, the limited data suggest a higher quantity taken in Africa (Table U in S1 Appendix). The probability of a predator (predatory mammals, birds of prey) being threatened by illegal hunting

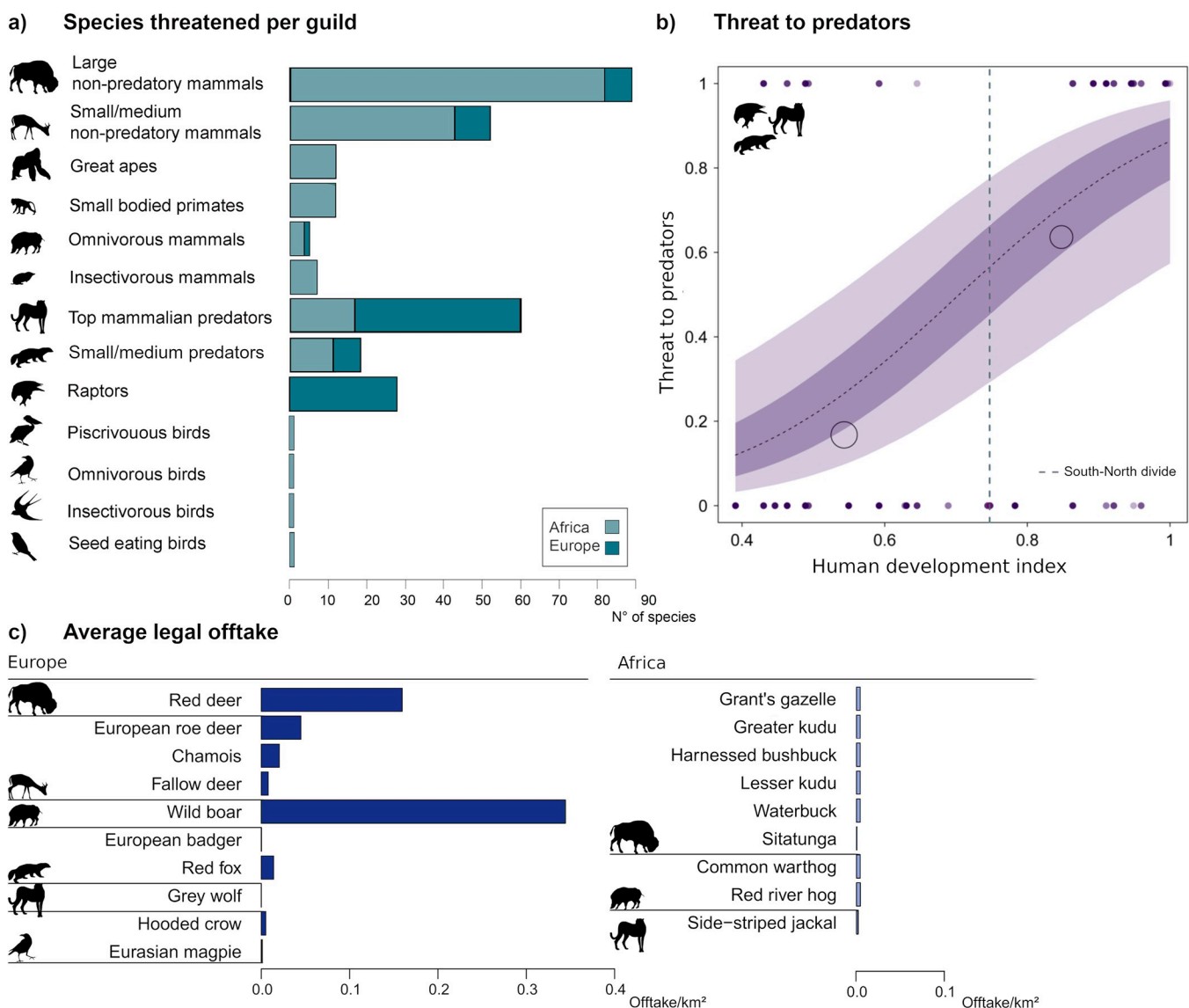

**Fig 3. What people hunt.** (a) The absolute number of threatened species per guild listed by park managers (includes repeated mentioning of the same species by different managers). In Africa, more herbivorous species, and in Europe, more top mammalian predators and raptors were named. Africa = violet, Europe = turquoise (the data underlying this figure can be found in S2 Data). (b) The probability that a predatory species (top, small to medium predators, raptors) compared to nonpredatory species (all others) were threatened showed an increasing trajectory over the S-N gradient (i.e., HDI) (S7 Data). The dashed line depicts the expected mean of the predicted posterior distribution. The coloured areas describe the 33% and 66% credibility intervals. Transparent points are binned probabilities. The size of the bubbles corresponds to the respective number of PAs. Filled points are original data. The vertical grey dashed line is the Global South-North divide of our data (0.75), (c) the legal offtake per species and square kilometre across European parks (left, turquoise) and across African parks (right, violet) (Table U in S1 Appendix and S3 Data). HDI, Human Development Index; PA, protected area.

increased with higher HDI scores compared to nonpredatory mammals and birds (Fig 3B and Table J in S1 Appendix, estimate = 1.22, CI = [−0.11, 2.36]). Predators were more likely listed as threatened by illegal hunting when their abundance increased over the last ten years (estimate = 0.57, CI = [−0.31, 1,49]).

## Where people hunt and how unsustainable hunting can be mitigated

In terms of where illegal hunting activities were spatially located, 63.1% of the species (130 of 206 cases, 56 distinct species) were threatened more when ranging outside borders, and 36.9%

(76 of 206 cases, 45 distinct species) were threatened more within the park territory (Fig 4A). How far species range beyond borders is here dependent on their ecology. The 21 species listed as threatened only outside of PAs were mainly raptors ($n$ = 7) and mammalian predators ($n$ = 7). In Europe, predators were particularly threatened outside PAs, while in Africa, herbivores were the most threatened guild, inside and outside of PAs (Fig 4A).

Our models revealed a declining threat inside parks by illegal hunting at an increasing rate when approaching HDI>0.75 (Europe) (Fig 4B and Table K in S1 Appendix, HDI: estimate = −1.44, CI = [−2.46, −0.46]). Threats to animals when ranging outside PAs also declined, albeit to a lesser extent (Fig 4C and Table L in S1 Appendix, HDI: estimate = −0.95, CI = [−1.94, −0.01]. The data distribution suggests that the absence of the threat from illegal hunting particularly increased with the HDI. Protection efforts, including ranger patrols, a permanent research station, and buffer zones, were weakly associated with a lower threat from illegal hunting within PAs (estimate = −0.58, CI = [−1.51, 0.28]). Supportive local community characteristics, including mutual trust and conservation-friendly attitudes and culture, were linked to lower threats from illegal hunting to animals within (Fig 5A and Table K in S1 Appendix, estimate = −0.99, CI = [−1.91, −0.12]) and outside of PAs (Fig 5E and Table L in S1 Appendix, estimate = −0.89, CI = [−1.83, 0.01]). When we included the components of the "local community characteristics" index separately in the models, mutual trust between park management and communities was linked to lower threat probabilities inside (Fig 5C and Table M in S1 Appendix, estimate = −1.26, CI = [−2.23, −0.34]) and outside of PAs (Fig 5G and Table N in S1 Appendix, estimate = −0,82, CI = [−1.73, 0.03]). Likewise, conservation-friendly attitudes reduced threat probabilities inside (Fig 5B and Table O in S1 Appendix, estimate = −0,82, CI = [−1.72, 0.02]) and outside of PAs (Fig 5H and Table P in S1 Appendix, estimate = −0,71, CI = [−1.72, 0.27]). Similar effects were observed for conservation-friendly local cultures both inside (Fig 5D and Table Q in S1 Appendix, estimate = −0,7, CI = [−1.6, 0.14]) and outside of parks (Fig 5H and Table R in S1 Appendix, estimate = −0,88, CI = [−1.82, −0.03]). In comparison to the effects of HDI, these effects were less pronounced within PAs than outside PAs. The models including only mutual trust best explained the results (Table S in S1 Appendix). When examining whether specific community interventions can foster such beneficial community conditions, the only predictor that showed an effect was "scale of inclusion of local communities" (Fig 5I and Table T in S1 Appendix, estimate = 1.4, CI = [−0.21, 2.77]) (alongside "provision of benefits to communities", "implementation of livelihood projects", and "awareness creation").

## Discussion

Although comparative studies are rare, hunting in the Global South is often portrayed as fundamentally different from hunting in the Global North [18]. Our results confirm differences in basic hunting characteristics (why, what, and where people hunt) across the studied Africa-Europe South-North gradient. Towards the north, the rate of economically motivated and highly unsustainable hunting that threatens herbivores both within and outside parks declines. Instead, the rate of legal and socially or ecologically justified hunting of ungulates increases, while threats to wildlife persist beyond park boundaries, albeit to lower extents. Threats to predators from illegal hunting increase along the S-N gradient. While some differences, such as the threat of hunting, varied gradually across the S-N gradient, others, such as the function of hunting or the species threatened, showed a clear separation between Africa and Europe. Notably, the commonalities found are mainly related to factors that mitigate illegal hunting, namely, favourable characteristics of local communities, including conservation-friendly attitudes and cultures, and, in particular, trusting relationships associated with higher levels of

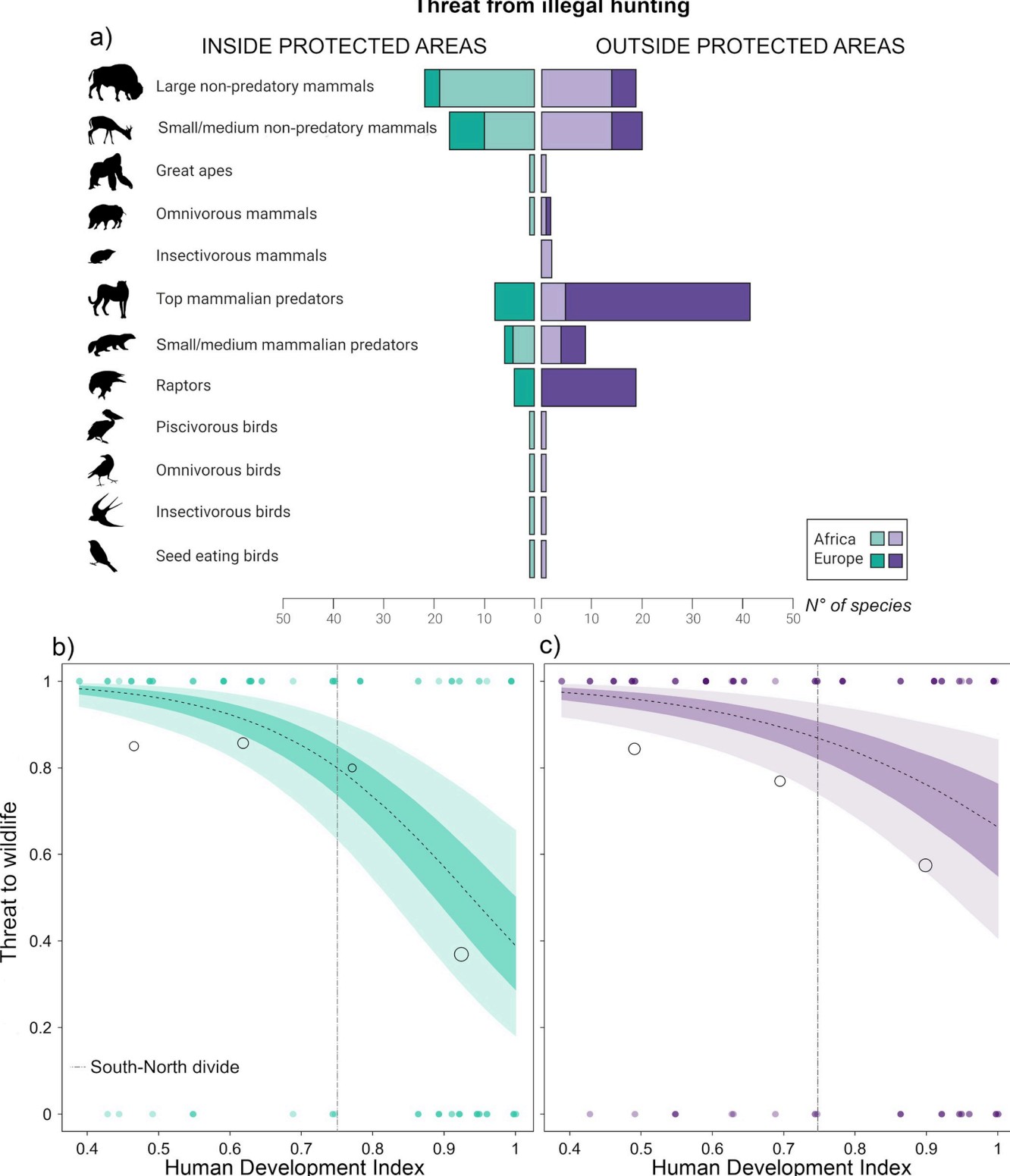

**Fig 4. Where people hunt.** (a) The absolute numbers of listed species of the PA threatened by illegal hunting within the administrative borders versus outside the park borders revealed higher threats for predators outside borders in Europe (darker colours) and higher threats for herbivores inside and outside parks in Africa (lighter colours). The probability that wildlife is threatened by illegal hunting (killing/hunting/poisoning) (the data underlying this figure can be found in S2 Data), (b) within the park boundaries (S8 Data) and (c) outside the park boundaries (S9 Data) decreased over the S-N gradient. Outside parks, the

probability decreased to a lower extent. The data distribution revealed a decreasing threat with higher HDI values. The dashed line depicts the expected mean of the predicted posterior distribution. The coloured areas depict the 33% and 66% credibility intervals. Transparent points are binned probabilities. The size of the bubbles corresponds to the respective number of PAs. Filled points are original data. The vertical grey dashed line is the Global South-North divide (HDI of 0.75). HDI, Human Development Index; PA, protected area.

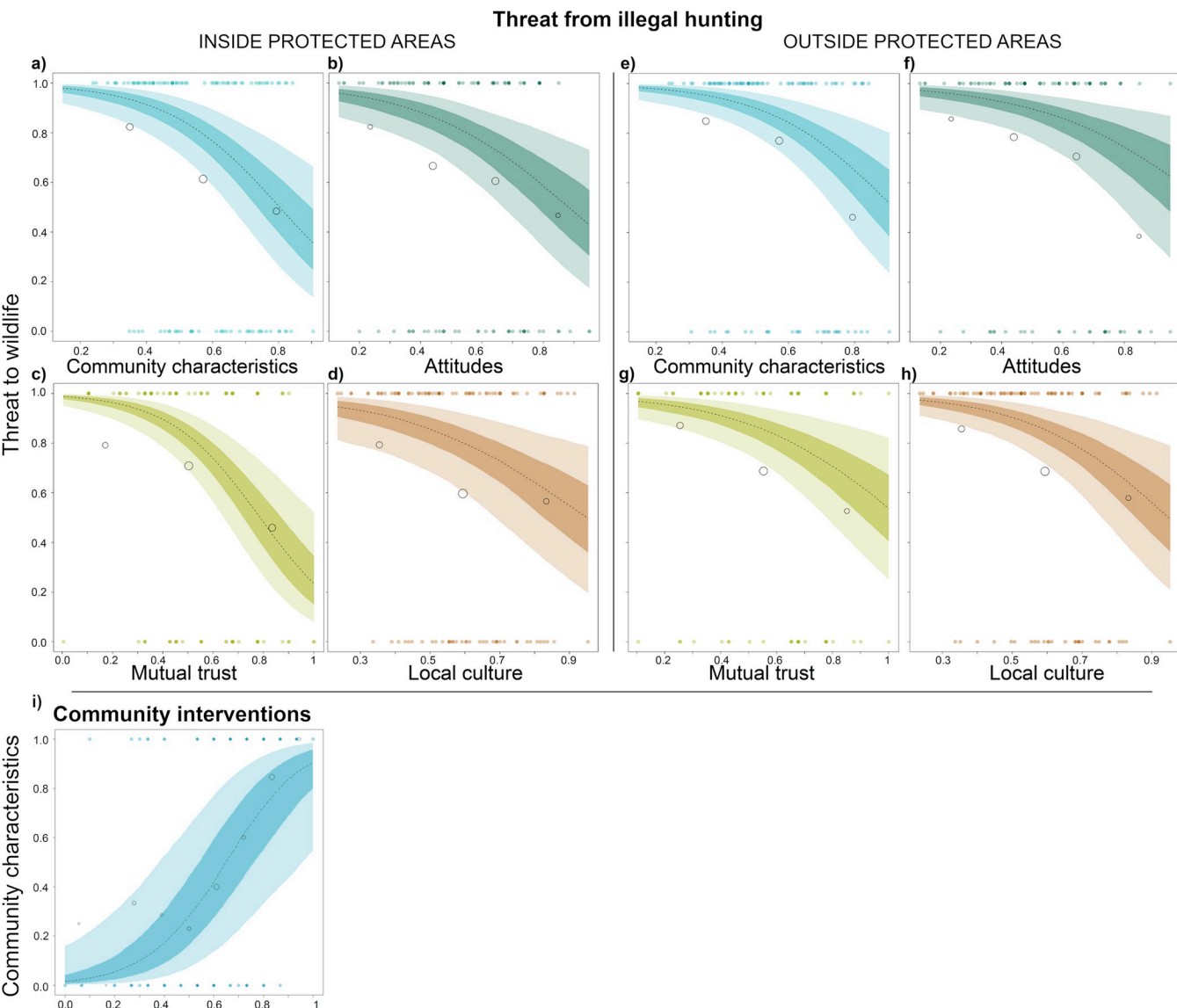

**Fig 5. Where people hunt and what mitigates unsustainable hunting.** Left panel: probability of high threat to wildlife by illegal hunting (killing/hunting/poisoning) inside PAs related to (a) community characteristics and the separate components of the index community characteristics, (b) attitudes of local communities towards conservation, (c) mutual trust levels between park management and organisations, and (d) conservation-friendly local culture (the data underlying this figure can be found in S8 Data). Right panel: threat levels outside park boundaries across (e) community attributes, (f) attitudes, (g) mutual trust levels, and (h) local culture (S9 Data). Threat levels decreased equally outside and inside parks, while the effects were more pronounced outside parks. The models, including mutual trust, best explained the results. (i) The scale of inclusion of local communities in decision-making showed a positive relationship with the index community characteristics (S10 Data). The "scale of inclusion" was alongside the "provision of benefits to communities", "implementation of livelihood projects", and "awareness creation", the only predictor that showed an effect. The dashed line depicts the expected mean of the predicted posterior distribution. The coloured areas are the 33% and 66% credibility intervals. Transparent points are binned probabilities. The size of the bubbles corresponds to the respective number of PAs. Filled points are original data.

community involvement in PA management. Nevertheless, these factors do not compensate for the strong South-North effect of the HDI. Last, conflict-driven hunting shows an increase with higher human population densities.

## Differences across the Africa-Europe South-North gradient

Considering why people hunt, we observed a prevailing economic function in Africa. This observation reflects the needs of those whose livelihoods depend on the consumption and trade of wild meat [9], despite such hunting not being permitted by national law in many countries. In Europe, we observed a division into an ecological function, which primarily reflects state-regulated wildlife control, and a social function, which reflects the recreational, social, and cultural motivations of hunters and their associations, with both functions permitted by legislation. The prevailing social function is unexpected because hunting in PAs is frequently presented as an ecological necessity [9]. In addition, ecological factors, such as the presence of large predators, show little effect on offtake rates. Some parks, however, have adapted to the return of large predators, such as the Czech Šumava National Park, which declared no-hunting zones after the arrival of a wolf pack [27]. Nevertheless, the prevailing social function might confirm that current ungulate management in Europe is determined by the sociopolitical contexts rather than by ecological parameters or the guidelines of the International Union for Conservation of Nature (IUCN) [28].

Our findings suggest that low HDI levels are a major cause of unsustainable hunting, which is mediated by the primary economic purpose of reliance on wildlife products for livelihoods [3]. Regarding hunting, uncontrolled and trade-driven hunting is the greatest threat to biodiversity [29], which increases the risk of disease transmission during sale, transport, and consumption [30]. Nevertheless, unsustainable hunting also occurs in Europe, albeit to a lesser extent, and partly involves legal hunting. Some managers reported a mismatch between authorities, regulations, and conservation and hunting objectives; e.g., in Slovakia and Hungary, only hunting associations set quotas in PAs, while in Austria, hunting laws prohibit the use of silencers near bird breeding areas (pers. comm. park managers, Table V in S1 Appendix). Furthermore, close collaboration between stakeholders involved in the management of PAs and adjacent areas can be extremely difficult, as the mindsets of interest groups (hunters, farmers, foresters, PA authorities) often differ widely (pers. comm. park managers). Overall, the continental differences in legal frameworks for regulating hunting within the same IUCN management categories are striking and attest to the presence of disputable double standards that are likely an important cause of the observed patterns. Potential explanations include colonial legacies, with African hunters being stigmatised, criminalised, and displaced from PAs [18,19]. Furthermore, international organisations have been strongly involved in African PA management, while local actors have been insufficiently included [16] in establishing strong institutional structures regulating hunting. Nevertheless, some countries, such as South Africa, have a long history of wildlife culling in national parks and have only recently changed their policies [31].

In terms of what people hunt, high rates of ungulates are legally hunted in Europe. Illegal hunting entails more predators in Europe and more herbivores in Africa. The latter might be related to a higher demand for wildlife products suitable for consumption and commercialization. Moreover, several vulnerable species, including large herbivores, e.g., the European bison, have historically been hunted to (local) extinction. In combination with a lack of economic interest for other species, this has resulted in impoverished but resilient species communities in Europe [13,26]. The greater endangerment of predators in Europe, typically victims of HWC, might contradict our prediction, as the costs of HWC are disproportionately higher in

the Global South [17]. However, the extent of conflict is rarely proportional to the actual damage because cultural and social perceptions of potential threats and the values associated with wildlife are decisive [8]. Tolerance towards large carnivores differs widely across Europe due to variation in cultural values (e.g., attitude towards wolves in Sweden and Norway) and the level of experience with coexistence over centuries [32]. In regions that are recolonised by large carnivores after long periods of absence, people tend to have more negative perceptions towards these carnivores compared to those who live in areas with continued coexistence [33]. Similarly, our results suggest that with growing populations, predators are increasingly targeted, presumably in areas where people have grown unaccustomed to their presence [5]. The social function of hunting in Europe can encourage the persecution of predators if such hunting is associated with enjoyment and status or competition over common prey [5,8]. Moreover, policy signals that allow, for example, the state-level culling of wolves can send negative messages about their conservation value and the acceptability of poaching, which can ultimately increase the poaching of these protected species [34].

Moreover, the ecological function of and thus the support for lethal wildlife controls are typically linked to higher wildlife value orientations of domination, which are widespread in Europe [35]. Here, human needs are strongly prioritised over the perceived needs of wildlife, while in a mutualist wildlife value orientation, both needs are considered to be equally important [36]. These value orientations can affect coexistence. The reintroduction of wolves was, for instance, more successful in the US states with prevailing mutualistic value orientations [36]. Our results suggest that illegal persecution is not severe enough to decimate the increasing predator populations, thereby confirming that predators are generally on the rise throughout Europe [4]. Nevertheless, single populations, such as lynxes (*Lynx lynx*) in the Bohemian Forest Ecosystem, are strongly endangered by illegal persecution [4,5].

Regarding where people hunt, our results show a high level of threats to wildlife from illegal hunting of any kind both inside and outside PAs in Africa and outside PAs in Europe. Alternative livelihoods with an increasing HDI and a much stronger regulatory framework outweigh the benefits of illegal hunting and render risky hunting inside PAs useless. Additionally, African PAs frequently lack the financial resources needed to engage in effective protection efforts inside parks [37], while outside parks, they are much weaker or even absent. Moreover, hunting concessions open to the trophy hunting of protected species often neighbour national parks [38]. For example, male lions from park interiors frequently reoccupy territories that have been emptied by trophy hunting outside the park, thereby precipitating declines across the whole area [39]. In Europe, PAs are also embedded in hunting areas, and they provide, for example, vital source areas for lynx kittens to mature safely; however, the illegal killings of dispersing subadults have a strong sink effect on the population [40]. These killings are often cryptic, with carcasses buried or animals poisoned, and the risks of being caught are low [41]. Migratory species and predators with large home ranges (i.e., 917 km$^2$ for male Eurasian lynxes [42]) are particularly affected. PA border areas frequently represent population sinks for predators, even though they enjoy mostly legal protection [5,40,43]. European parks are too small to protect large carnivores and other large mammals effectively. They may represent safe zones (for hibernating, pup and kitten rearing, calving, wintering), but most carnivores and large herbivore species have home ranges that are much larger than typical European national parks. Consequently, predators largely exist outside European PAs [4,40] and are likely to disappear from small reserves, irrespective of their population size, when the hunting pressure is too high [43].

## Commonalities across the Africa-Europe South-North gradient

Despite all the historic, socioeconomic, and environmental contrasts, commonalities do exist. Across the studied PAs, human density is associated with increased killings due to HWC. This finding may be a simple function of increased encounters and competition between wildlife and growing numbers of humans. Likewise, wildlife adapt their behaviours when fewer habitats or natural dietary items are available, which can increase the level of depredation on livestock [44]. This link might be archetypical, since over millennia, growing populations of humans and their livestock have replaced wild animal populations [26]. However, we found no effect of human densities on predators. Indeed, all 4 large European carnivore species, namely, the brown bear (*Ursus arctos*), lynx, wolf, and wolverine (*Gulo gulo*), persist in human-dominated landscapes (i.e., wolves: $36.7 \pm 95.5$ inhabitants/km$^2$) [4]. Coordinated legislation across Europe, context-specific management practices, and institutional arrangements have enabled this conservation success [4,7], thereby ultimately outweighing this archetypal human–wildlife relationship.

Considering the question of how unsustainable hunting can be mitigated, we confirmed an effect of protection-based conservation that decreases threats to wildlife from illegal hunting inside PAs to some (modest) extent. This low effect of conservation measures may suggest that PAs are increasingly struggling to protect biodiversity from the pressures exerted by the larger context [2]. Alternatively, our findings could reflect reversed causality, in which more interventions are applied when wildlife is already threatened and the effects of threat reduction are difficult to detect [45].

Regarding what mitigates the threats to wildlife from illegal hunting inside and, in particular, outside parks, another identified commonality is the effect of supportive local community conditions. After conservation-friendly attitudes and cultures, mutual trust between management and communities showed the strongest effect. Trust has proven to be central to biodiversity conservation [46], including the success of PAs [25] and carnivore protection [47]. Currently, widespread and frequent mistrust remains among stakeholders (hunters, farmers, foresters, conservation managers, societal actors) [48]. Trust can be broken when stakeholders feel that other parties' interests are being promoted at their expense [46]. This typically correlates with varying perceptions, uses, priorities, and impacts of wildlife or existing power gradients, such as the rural-urban gradient or the Global South-North gradient, or strongly divergent traditions in institutional settings and results in in-group versus out-group dynamics [14,46–48]. Important trust-building mechanisms consist of transparent and fair decision-making processes and the ensured participation of all parties [46,49]. Notably, the inclusion of local communities in management contributes to supportive community conditions. Externally imposed rules can lead to animosity towards conservation or forms of "resistance poaching" that purposely target key conservation species [10,14]. In Europe, hunters could historically largely act independently in their concession, and the increasing influence of conservation bodies and the return of predators seem to jeopardize this exclusive control over huntable wildlife. In sub-Saharan Africa, however, and among some conservation agents, exclusionary and militarised approaches are becoming the norm, which arouses deep levels of mistrust among residents [14,18]. Conversely, well-implemented inclusive approaches benefit the social, economic, and ecological outcomes of PAs [25] by ensuring equity and trust, reinforcing positive attitudes, and considering the plurality of cultural values associated with nature [50].

## Potential regional and universal solutions

Overall, various manifestations and impacts of hunting on wildlife highlight the need to move beyond oversimplified notions [51]. The differences we observed help in the derivation of

concrete practical solutions in the short to medium term. The improvement of living conditions and human well-being in low-HDI countries are likely prerequisites for reducing unsustainable hunting. However, unsustainable hunting occurs along the entire S-N gradient and the prosecution and conviction of illegal hunting activities is rare, both in Africa and Europe [41,52]. Too often, the illegal killing of wildlife is treated as a trivial offence, and cases are quickly terminated without sentencing the suspected person, thereby potentially encouraging others (pers. comm. park managers). Recognising poaching as a serious problem in Africa and Europe, the training of police, and enhanced collaboration among conservation bodies and law enforcement agencies to actively prosecute wildlife crime are important steps forward. In the Bavarian Forest NP, for example, national park staff train police to secure evidence of illegal killing of lynx (pers. comm. Marco Heurich).

Additionally, as observed in our study, hunting in European PAs serves not only an ecological function but also a social function (Fig 2A). This is disputable and emphasises the call for an integrated European management policy [28]. PA management needs to obtain the main authority over hunting activities in PAs to develop management plans in cooperation with hunters so that hunting activities fully serve the PAs' objectives. Conversely, in African PAs, limited evidence on the social function of legal hunting for local people exists, which likely results in the fulfilment of this function through poaching and conflict with conservation bodies. Moreover, the continued hunting, even in large quantities, in European PAs reflect questionable double standards between Europe and Africa. Conservation science and practices should address these and discuss sustainable use by local people to be able to tackle biodiversity crises.

Likewise, some European countries seem to struggle to protect predators, control their populations within PAs (e.g., Sweden), and oppose rewilding, while simultaneously advocating for human–wildlife coexistence abroad with costs disproportionately borne by rural populations in the Global South [17]. These inequalities undermine the credibility of universal wildlife conservation and human–wildlife coexistence efforts and standards in PA management. Acknowledging these double standards while seriously considering the demands of inhabitants and empowering the decolonisation movements [16] would be important steps towards eventually overcoming them.

Even though HWC increases with human population densities, the growing carnivore populations in Europe show that good institutional and governance settings can change the archetypal link between human-dominated landscapes and dwindling wildlife populations and enable coexistence.

However, how can these good conditions be created? The identified commonalities that mitigate harmful hunting could help us understand universal mechanisms [20] and thus provide conditions in social-ecological systems in which people and wildlife could thrive. Given the contrasting socioeconomic conditions along the S-N gradient in which these commonalities occur, they may be explained by ultimate causes related to general human behaviour and our history as a social species [49,53]. Ostrom and colleagues identified a set of minimal conditions in social-ecological systems, also core design principles, that root in evolutionary aspects of human cooperation and must be met to enable sustainable resource use [49,53]. Mutual trust, which is a key attribute of sustainability as identified in our study, is essential in any setting where parties must renounce immediate individual benefits for the long-term good of the whole [49]. Trust is, however, also an emergent property under good governance, inclusion, and functional arrangements [49]. This includes making activities and processes transparent, stakeholders taking responsibility for their action and being held accountable for what they do [49] and overcoming in-group favouritism and the associated diverging mindsets [48]. Concrete solutions for achieving these aims for sustainable hunting practices include considerably

improved communication and exchanges among stakeholders that can be fostered and mediated by the cross-site engagement of wildlife experts and professional mediators [54]. Moreover, functioning mechanisms to enforce commonly agreed-upon regulations, preferably through graduated sanctions, which reach from social pressure to law enforcement are needed [49]. In the vicinity of the Bavarian Forest National Park, for example, the continuous illegal hunting of lynx declined after there was public outcry leading to substantial law enforcement following the killing of "Tessa," a prominent lynx in the region, who was followed by the media for a year as she raised her cubs (pers. comm. Marco Heurich).

Furthermore, as another key finding, the inclusion of all relevant group members to jointly organise activities, create rules, make decisions, and find solutions is vital for collaboration [49,53]. Inclusive approaches foster mutual trust, enable local adaptation, and trigger intrinsic incentives since people resent being told what to do but work hard to meet agreed-upon goals [49,53]. Round tables that include stakeholders and societal actors allow constructive debates about potentially diverging perspectives on human–wildlife coexistence and HWC; learning about the viewpoints of other interest groups is an important activity for achieving this goal [55]. However, the feasibility of inclusive approaches depends on a range of factors, such as the existing local structures, the process design, facilitators, the history of conflict and trust, or the capacities and interests of the participants (pers. comm. park managers) (e.g., [56]). Although the design principles can be specified in general terms, their specific implementation requires a process of local adaptation [49,53]. Crucial here are consensus decisions that prevent some group members from imposing decisions at the expense of others [49,53]. Nevertheless, frequently, important stakeholder groups or societal actors are not part of important decision-making processes either because they are not identified as stakeholders or intentionally excluded or because asymmetries exist between stakeholders, e.g., economic versus environmental interests, between rural-urban or the Global North and South or with regard to cost and benefit sharing (pers. comm. park managers) [57]. In citizens' assemblies, which is an increasingly used tool for addressing polarised issues, randomly selected citizens develop informed policy recommendations based on consensus decisions after input by stakeholder groups and scientists. Usually, the environmental policies agreed upon in these situations surpass the existing level of ambition by governments for solving urgent challenges [58]. In the rural Global South, informal and community-oriented approaches typically provide the context for efforts to promote coexistence with wildlife [21].

The threats that affect wildlife beyond PA boundaries may show conflicts that occur because people and wildlife follow different system boundaries [21]. Core design principles of sustainable systems, however, require that the boundaries around the community of users and the resource itself must be clearly delineated [49,53]. Stronger social boundaries through shared values, rules, or goals can outweigh fuzzy geographic and political boundaries that are not respected by mobile wildlife [59], which might explain why beneficial community conditions can reduce the hunting threat to wildlife, particularly beyond borders. This requires finding common ground in hunting and natural resource use among involved stakeholders and identifying overlaps in objectives, such as the control of invasive species. Establishing positive attitudes towards wildlife and conservation and nature-friendly cultures can be achieved by political and societal discourses that aim at fostering human–wildlife coexistence, i.e., turning away from a strongly pronounced people-first mindset and turning towards mutualist wildlife value orientations [60]. This requires, however, continued and substantial investment from wildlife and governmental authorities into sensitization and awareness campaigns, particularly in areas where people have become unaccustomed to the presence of large predators and herbivores. Finally, parks are largely influenced by factors beyond the control of managers (i.e., HDI, population densities, threat beyond borders). This underscores a general shortcoming of

the PA concept, as parks are not closed systems and parts of larger systems that require collaboration across multiple scales and levels [53,61]. Decision and policy makers must be made aware of the potential consequences of their actions for wildlife, even when those actions are apparently not directly linked to PAs. For example, recent changes in wolf hunting policy outside Yellowstone National Park led to a 20% decline in the wolf population within the park [62]. Therefore, we need functionally connected PAs, i.e., through PA networks, which are adapted to species' ecologies, complementary protection efforts outside PAs, and proactive strategies that enable coexistence, as the survival of returning wildlife in Europe will ultimately depend on people's willingness to share landscapes and resources [7,63].

Critically, some arrangements might violate core principles, such as equal cost–benefit sharing, self-determination, and inclusion, i.e., apparent continental differences in legal settings and the absence of social function of hunting in Africa, which might indicate lacking local participation in wildlife management and unequal costs and benefits from wildlife. These examples can poison the collective efforts needed for wildlife protection [53]. More drastically, they reflect demands for decolonisation and environmental justice, thus entailing a fair distribution of costs and benefits, participation in decision-making and the recognition of identities and cultures [64]. Since traditional PA management may have a low compliance level with core design principles [65], linking the core design principles with local knowledge and values could benefit conservation work and just arrangements.

## Limitations of the study

Similar to most PA assessments [2,61], our data reflect the perceptions of PA managers, which can involve certain biases and can differ from the perceptions of other stakeholders [9]. As our findings are solely based on correlational analyses rather than experimental work, we would like to point out that all the subsequent interpretations of the results have this same limitation. The associations we identified could also be interpreted in different ways. To improve our understanding of the role of the core processes in wildlife hunting highlighted in this study, further knowledge synthesis across diverse gradients of actors, institutional arrangements, and societies is needed. Moreover, this study examines only 1 exemplary Global South-North gradient between 2 continents, Africa and Europe. Therefore, more research including other continents is needed to understand the gradual changes and commonalities across the South-North axis. It is also important to note that the HDI used to estimate the current S-N gradient (slope) and correlations with the HDI are not necessarily causations and might instead be a proxy for different historical, biodiversity, legal, or other contextual settings.

## Conclusions

Along the entire Africa-Europe South-North gradient, various forms of hunting, such as socially, economically, or ecologically motivated illegal and legal hunting and HWC, pose a major threat to wildlife both in and around PAs. Future trajectories will likely worsen the identified wildlife-related challenges. Climate change and biodiversity loss will exacerbate the historical inequalities that are currently driving the unsustainable hunting associated with biodiversity and disease risks in the Global South [30,66]. These changes will further push people into new areas, thereby increasing the level of HWC and the demand for natural resources due to population increases and aspirations of perpetual economic growth [21]. Current PA management, either control based or exclusionary, will be called into question by increasing demands for codetermination in the South and growing predator populations in the North, both of which challenge human–wildlife relationships. Beneficial local community conditions have protective effects on wildlife but cannot reverse the strong South-North effect. Consensus

is growing that only systemic changes in economics, values, and society can prevent biodiversity loss and address long-standing inequalities [50,66]. PAs and local communities cannot protect their wildlife until large-scale and sustainable economic development removes the economic constraints that currently drive poachers to exploit even remote parks. Wildlife will only be tolerated in human-dominated landscapes in the Global North when people accept that their rights mutually exist. These challenges entail social and normative negotiation processes at every single organisational level and the development of context-specific adaptive practical solutions and arrangements, which require strong institutional settings on different levels. Theory suggests that the minimal core principles briefly outlined above are applicable across contexts and scales and empower every relevant level to resolve within- and between-level conflicts, thereby ultimately enabling more networked, just, and effective wildlife conservation [49,53].

## Methods

### Data collection

We collected the data through face-to-face interviews with managers of 114 African and European PAs using a structured questionnaire [2,61] (Fig 1A and Table A in S1 Appendix) (see the questionnaire in S2 Appendix). We obtained the polygons of PAs from the World Database of Protected Areas [67]. The study was approved by the Ethics Council of the Max Planck Society, and we obtained informed consent from all participants. We used 3 main criteria to sample the PAs from the World Database of Protected Areas [67]: (1) if possible, PAs in the category of national parks; (2) the availability of data to cross-validate our questionnaire data (Living Planet Database (LPD) [68], IUCN SSC A.P.E.S Database [69] reports) (Table W in S1 Appendix); and (3) permission from governments to conduct surveys and the willingness of PA managers to complete our questionnaire. We used face-to-face interviews to circumvent the challenge of acute data shortages on hunting and conservation strategies from an ecological, economic, and social perspective [2] (Questionnaire can be found in the S2 Appendix). As we were interested in the interface between biodiversity protection and resource use, we selected PA directors or their representatives as our interview partners. We expected the highest data availability and expertise on hunting and its impacts on biodiversity from these key stakeholders [2]. We conducted the survey from December 2017 to September 2019 in 25 African and European countries and 114 parks (Fig 1A and Table A in S1 Appendix). We surveyed the southern, eastern, western, central, and northern parts of Europe and Africa to capture a steep yet nuanced gradient regarding socioeconomic and PA management conditions along the Global South-North axis. This gradient ranges from countries in Central Africa with extremely low HDI values to the high HDI-scored countries of Northern Europe but also countries in Southern and Northern Africa that reach values similar to those in Eastern Europe (Fig 1A). We used standardised questionnaires to collect data on socioeconomic dimensions, community attributes, aspects of hunting, and conservation interventions (see questionnaires in S2 Appendix). We further asked about changes in mammal and bird abundances over the past 10 years (overall 462 species, 15 functional guilds/groups). Given that data of all occurring species were not available, the listed species are those where PA managers were able to report changes. Moreover, we collected grey literature in the form of reports and ecological and socioeconomic surveys conducted by parks, management plans, lists of confiscated animals, and records of confirmed poaching cases (Table U in S1 Appendix). Even though we consider PA managers the best sources of this information, responses can deviate from reality due to unconscious and conscious biases. Thus, we cross-checked the validity of our data with data collated from the LPD [68], IUCN SSC A.P.E.S Database [69], published and unpublished

reports, and Protected Area Management Effectiveness (PAME) [70] assessments. Furthermore, we conducted an online survey with the same questionnaire with nongovernmental organisations (NGOs) working in the surveyed PAs. We grouped similar variables as in our analyses into 2 levels (declined or stable/improved and low, high) and compared the values between our data and these reference sources. For our abundance data, we found an overlap regarding increasing, stable, or decreasing population trends in 82.4% of the cases compared to other sources (Table W in S1 Appendix). When comparing questionnaires completed by NGOs and PA managers, we found an average overlap of 60.04% ± 12.27%. For the PAME assessment, we found an overlap of 54.8% ± 13.3%. However, the power of the comparison was limited due to the low number of responses for the NGO survey and for the PAME due to the low number of comparable questions and different wording and time periods. Further details on this cross-validation are included in section 4 of the S1 Appendix.

## Models

We implemented Bayesian hierarchical regression models (BHRMs) using the "brms" package [71] in R. As priors, we used a standard normal distribution with a mean and standard deviation of 1. We initially performed 2,000 iterations over 4 Markov chain Monte Carlo (MCMC) chains but increased it due to failure in some models to 10,000 iterations [71]. In all models, we controlled for spatial autocorrelation by including a Gaussian process over longitude and latitude for each PA [72,73] by using the function "gp" from the "brms" package [71]. We standardised all variables from 0 to 1 before compiling the indices. All predictor variables were transformed into a standard normal z distribution (with mean 0 and standard deviation 1) to facilitate comparison of the results from our models [74]. We tested and ruled out correlation and multicollinearity among predictors. Statistical analysis was conducted using R 3.6.1 [75]. The inspection of the MCMC results showed stationarity and convergence to a common target. All "Rhat" [76] values were below 1.01, and we had no divergent transitions after warmup. As we lacked prior information, we ran the models with weakly informative priors with a standard normal distribution (mean of 0 and standard deviation of 1). To achieve roughly symmetrical distributions and avoid influential cases, we square root- or log-transformed skewed predictors.

## Responses

To understand the differences and commonalities in hunting characteristics across the S-N gradient, we used 7 binary response variables in 9 models (Table 1).

i. Function of hunting: First, we investigated the multiple functions of hunting, defined as providing goods and services [9]. We grouped these based on their prevailing motivations into the three-dimensional structure common in sustainability science (e.g., [9]). An economic function combined hunting for subsistence and commercial interests. A social function comprised nonmarket-related hunting for entertainment and cultural and social interests, and an ecological function included population control and killing due to HWC [9]. The response was an overall hunting index, reflecting whether a function is one of the main motivations or only provides a negligible motivation for hunting in the park. We tested the effect of single functions by an interaction between the category of the hunting functions (economic, social, ecological) and all main predictors (HDI, population density, community attributes).

ii. Manifestations of wildlife interactions: Second, we explored the prevalence of 2 common human–wildlife interactions. We used as a response the threat assessments of "hunting"

and "killings because of human–wildlife conflicts". We included legal and illegal actions but explored whether including only illegal hunting would change the results.

iii. Trophic level: Third, we assessed differences in hunting pressure between trophic levels of target animals. Our response was predator versus nonpredators threatened by hunting as listed by managers. We defined small to large predatory mammals and birds of prey as predators, while primates, apes, insectivorous, omnivorous, and herbivorous mammals and birds were considered nonpredatory.

iv. Threat location: Fourth, we aimed to understand differences in the spatial location of illegal hunting. Our first 2 responses were a rating of how protected species from the park were affected by illegal hunting within administrative park borders or when ranging outside the park borders. We included the components of our community characteristics attitudes, trust, and local culture individually and derived the Akaike information criterion (AIC) values from the differences between the models (ΔAIC) [77].

v. Follow-up analyses: We carried out follow-up analyses based on our results to further understand the scope of actions available to park managers. We tested whether individual interventions of the index "community-based interventions" can foster the supportive local community characteristics that have been shown to protect wildlife inside and outside parks. The response was the index for "community characteristics". Predictors were the index components of "community conservation effort", namely, "provision of economic benefits to the community", "implementation of livelihood projects", "scale of local inclusion", and "implementation of environmental awareness programs", separately in the model. In addition, the model contained population density, size, and country as control predictors and used "continent" instead of HDI due to correlations.

Construction of indices and binary responses:

All questions were recorded on ordinal or Likert scales. We compiled indices in the following steps: (1) We log- or squared-transformed skewed variables to ensure normal distribution and to enhance comparability. (2) Since variables were collected on different scales, we standardised all variables to a range from 0 to 1 to enhance interpretation. (3) When variables were collected on opposite scales, we reversed variables to equalise the interpretation direction. (4) Finally, we derived an index by summing the single variables over the mean. We transformed the responses into binary responses, reducing the scope of error [78]. We controlled for odd numbers if a different break changed outcomes (i.e., grouping questionnaire answers moderate, high, very high, or, respectively high or very high as high threat (Tables E–I in S1 Appendix) or breaks at <0,5 versus > = 0.5 (Tables C and D in S1 Appendix). For an overview of responses, see Table 1. For further details about the construction of responses and indices, see Table B in S1 Appendix.

## Predictors

We approximated the S-N gradient with the HDI (2017), a composite national index of life expectancy, education, and per capita income (gross national income GNI (PPP) per capita) [79]. We included multiple-scale predictors known to affect human–wildlife relationships. At the landscape scale, we included human population density (2015) [2] (Fig A in S1 Appendix). At the local scale, we constructed an index for community characteristics based on questions regarding nature–culture relationships, attitudes towards nature, conservation, the concept of PAs and their current management, and levels of mutual trust between communities and park management [25] (Fig B in S1 Appendix). To account for conservation efforts, we generated 1

index for protection-based interventions that protect the resources within the park. The index included the presence of regular ranger patrols, buffer zones, and a permanent research station, and even if the primary intention is different, the presence of staff can have a similar protective effect as ranger patrols [80]. We further compiled 1 index for community-based interventions, including conservation efforts altering the local context beyond park boundaries through community-based interventions (provision of benefits to communities, implementation of livelihood projects, awareness creation, inclusion of local communities). Which municipalities and in which area belong to the local communities was determined here by the park manager. We controlled for park size [61] and included the country as a random effect. We included all necessary random slope components. For the construction of the variables, see Tables A and B and C in S1 Appendix. We included all index components separately in the model to test the robustness of our indices.

## Supporting information

**S1 Appendix. Survey and results.**
(DOCX)

**S2 Appendix. Questionnaire.**
(PDF)

**S1 Data. Data Fig 1A and 1D.**
(XLSX)

**S2 Data. Data Figs 1B, 3A, and 4A.**
(XLSX)

**S3 Data. Data Figs 1C and 3C.**
(XLSX)

**S4 Data. Data Fig 2A.**
(XLSX)

**S5 Data. Data Fig 2B.**
(XLSX)

**S6 Data. Data Fig 2C.**
(XLSX)

**S7 Data. Data Fig 3B.**
(XLSX)

**S8 Data. Data Figs 4B, 5A, 5B, 5C and 5D.**
(XLSX)

**S9 Data. Data Figs 4C, 5E, 5F, 5G and 5H.**
(XLSX)

**S10 Data. Data Fig 5I.**
(XLSX)

**S1 Script. Analyses i.**
(R)

**S2 Script. Analyses ii.**
(R)

**S3 Script. Analyses iii.**
(R)

**S4 Script. Analyses iv.**
(R)

**S5 Script. Analyses v.**
(R)

## Acknowledgments

Our gratitude goes to all PA managers and staff who participated in our survey and completed our questionnaire (full list is given in Table A in S1 Appendix). Furthermore, we thank all who participated in the NGO survey: Zuzana Záborská (Regional Tourism Organisation Slovenský raj & Spiš), Karol Kaliský (Arolla Film), LZ VLK, Viliam Bartuš (WOLF Forest Protection Movement, Eastern Carpathians tribe), Hnutí DUHA Olomouc, Tomasz Pezold Kneževi (WWF Poland, IUCN WCPA), NHF, O. Ionescu (Transylvania University), Florin Stoican (Asociatia Kogayon), Andrei Szabo (Asociatia Euroland Banat), Asociatia Salvati flora si fauna Deltei Dunarii, Propark-Fundatia pentru Arii Protejate, Joseph Kouassi, Yves Kablan, Emmanuel Danquah (Department of Wildlife and Range Management, FRNR, KNUST, Kumasi, Ghana), Angedakin Samuel, and the CTPH Conservation Through Public Health. We would also like to thank Adam Bohdan, Sarah Bunel, Hayfe Chamkhi, Martina Duskova, Vidrige Kandza, Elysée Mbaygone, Nyakoojo Moses, Terence Fuh Neba, Peter Sabo, Clement Tweh, Andrada Vaidos, Mercy Wambui, and others for their support in collecting the data.

## Author Contributions

**Conceptualization:** Mona Estrella Bachmann, Lars Kulik, Tsegaye Gatiso, Martin Reinhardt Nielsen, Dagmar Haase, Marco Heurich, Dustin Eirdosh, Karsten Wesche, Hjalmar S. Kühl.

**Data curation:** Mona Estrella Bachmann, Lars Kulik, Tsegaye Gatiso, Isabel Ordaz-Németh, Tenekwetche Sop.

**Formal analysis:** Mona Estrella Bachmann, Lars Kulik.

**Funding acquisition:** Mona Estrella Bachmann, Karsten Wesche.

**Investigation:** Mona Estrella Bachmann.

**Methodology:** Mona Estrella Bachmann, Lukas Bösch, Hjalmar S. Kühl.

**Project administration:** Mona Estrella Bachmann.

**Supervision:** Hjalmar S. Kühl.

**Visualization:** Mona Estrella Bachmann.

**Writing – original draft:** Mona Estrella Bachmann.

**Writing – review & editing:** Mona Estrella Bachmann, Tsegaye Gatiso, Martin Reinhardt Nielsen, Dagmar Haase, Marco Heurich, Ana Buchadas, Lukas Bösch, Dustin Eirdosh, Andreas Freytag, Jonas Geldmann, Arash Ghoddousi, Thurston Cleveland Hicks, Isabel Ordaz-Németh, Siyu Qin, Tenekwetche Sop, Suzanne van Beeck Calkoen, Karsten Wesche, Hjalmar S. Kühl.

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
