## [Editor Report · Decision Letter 0]

23 Jun 2021

Dear Dr Bachmann, 

Thank you for submitting your manuscript entitled "Differences and commonalities in wildlife hunting across the Global South-North gradient" for consideration as a Research Article by PLOS Biology.

Your manuscript has now been evaluated by the PLOS Biology editorial staff, as well as by an academic editor with relevant expertise, and I'm writing to let you know that we would like to send your submission out for external peer review.

Please re-submit your manuscript within two working days, i.e. by Jun 25 2021 11:59PM.

Kind regards,

Roli Roberts

Roland Roberts

Senior Editor

PLOS Biology

rroberts@plos.org

---

## [Decision Letter · Decision Letter 1]

15 Sep 2021

Dear Dr Bachmann,

Thank you for submitting your manuscript "Differences and commonalities in wildlife hunting across the Global South-North gradient" for consideration as a Research Article at PLOS Biology. Your manuscript has been evaluated by the PLOS Biology editors, an Academic Editor with relevant expertise, and by two independent reviewers. We had recruited a third reviewer, but they were unable to submit in a timely manner, so we proceeded to a decision without their input.

You’ll see that both of the reviewers are very positive about the study overall; however, they each raise a number of concerns that will need to be addressed. Both have questions about the choice of variables and other aspects of the method, and both want some degree of caveating. Rev #1 wonders if you could incorporate some open-ended material from the park rangers, while reviewer #2 attaches a version of the PDF that has been annotated with helpful comments; s/he would also like you to avoid the use of the word “gradient” and to tell us more about the implications for practical solutions.

In light of the reviews (below), we will not be able to accept the current version of the manuscript, but we would welcome re-submission of a much-revised version that takes into account the reviewers' comments. We cannot make any decision about publication until we have seen the revised manuscript and your response to the reviewers' comments. Your revised manuscript is also likely to be sent for further evaluation by the reviewers.

We expect to receive your revised manuscript within 3 months. 

**IMPORTANT - SUBMITTING YOUR REVISION**

*Re-submission Checklist*

*Published Peer Review*

*PLOS Data Policy*

*Blot and Gel Data Policy*

Sincerely,

Roli Roberts

Roland Roberts

Senior Editor

PLOS Biology

rroberts@plos.org

REVIEWERS' COMMENTS:

Reviewer #1:

[identifies himself as Richard Fuller]

 GENERAL COMMENTS

This is an enormous study, conducting face-to-face interviews with park managers across 114 protected areas in 25 countries in Africa and Europe. The sheer breadth and volume of data make this a landmark study. I buy the main thesis of the paper, that unsustainable exploitation of wildlife is often seen as a problem peculiar to the global south, hampering identification of similarities and differences in patterns that can tell us something about the underlying processes.

The main body of the paper is brilliantly written, with great figures and text that concisely and clearly tell the unfolding story of the analyses. The results are compelling, and wonderfully unify hunting as a phenomenon that has interesting commonalities and interesting differences across a very broad gradient of socio-economic contexts. The discussion, however. goes quite some distance beyond the data, but by and large it feels OK. The one thing that does seem unfortunately missing from the discussion is the voices of the park managers themselves. Were there any open-ended questions in the survey that could be used to liven up the discussion with some thoughts from the front line?

[the questionnaire itself was not included in the version of the supplementary information that I could access, which was unfortunate]

However, my main concern is focused more "under the hood". Most notably, the way in which variables were constructed is unclear, hampering detailed interpretation of the results, and rendering the study unreproducible. For example, page 17 "Responses" is very confusing, with 7 binary response variables, 9 models, 5 bullet points, and only vague explanations of the response variables in each. For all of the response variables it is not clear exactly how they are calculated, or how they are binary as specified in the preamble to this section. This section needs to make clear to the reader precisely how the data from the questionnaire are processed to form the response variables. Many of the variables were derived from Likert scales, but how were data from multiple scales combined, and how were cutoffs for binary divisions determined etc. There do not appear to be any sensitivity analyses presented that explore different ways of deriving the variables. Even wading through Table S2 doesn't provide enough information for the reader to work out exactly how the responses in the questionnaire were used to calculate variable values. The same is true for the predictor variables, and much more clarity is needed around describing how all the variables used in the study were put together.

SPECIFIC COMMENTS

Page 4, para 1

Paragraphs starts well, but then goes rather hyperbolic, e.g. "synthesis along a diverse gradient may make it possible to identify deeper underlying mechanisms that outline generalizable pathways within the exceedingly complex phenomenon of wildlife overexploitation" - consider simplifying the writing in this paragraph to make it more direct and meaningful.

Page 9

Important to caveat many of these analysis as being correlational. They are not robust tests, e.g. of effects of protection efforts on threat. A robust test would require before / after, or BACI designs. I think the analyses are valuable, but there just needs to clear recognition that these are associative results only, and do not indicate effects of predictor variables on response variables. In addition there could be feedback effects, e.g. more ranger patrols being instituted when hunting levels are high.

Page 16, para 1

Unclear sentence: "For our abundance data, we found an overlap in 82.4% of the cases when comparing to other sources (S22 Table)". Unclear how overlap is judged. I appreciate S22 and section 4 of SI is being referred to, but this point should be made much clearer in the main text otherwise that prime real estate is being wasted. The very next sentence "When comparing questionnaires filled by NGOs and PA managers..." suffers from exactly the same problem.

Reviewer #2:

[please also see attached file]

Dear authors, I commend you for a comprehensive and well-executed study comparing wildlife hunting inside and outside PA's across 2 continents. The authors find commonalities and differences in the threat posed by hunting to wildlife and explore a range of predicted variables in an effort to understand the social, governance and economic aspects underpinning the patterns.

Please find detailed comments in the attached documents, but here are some major comments:

- Methods are suitable for the questions asked, but I would advise the authors to develop the choice of predictors so it aligns with the scope of the work; use literature to support the selection of predictors at various scales, and how do they relate directly to the questions posed here. It would also help if hypotheses are postulated based on these arguments.

- "Gradient": this was a major sticking point: I understand that he authors wan to sell this manuscript to a high IF journal, and that the "Global N-S gradient" has a certain allure to it, but this is comparison of European and African wildlife hunting in 25 PAs, no a comprehensive global analysis that warrants this name. In fact, the Results compare Europe and Africa, and the Discussion build on these comparison, and I can't see the 'gradient' anywhere. Also, the Introduction does not make ha case hat a gradient exists, but that comparative analyses of wildlife management systems in different continents can lead to improved inference compared to focusing on a single continent. It seems like the authors use it as a catchphrase to increase the appeal. If so, I strongly advise the authors to reframe the paper and be truthful to what this paper presents: a comparative study of hunting in 25 European and African PAs. This would not affect the outcomes or importance of this work. 

- The Discussions make a lot of unsupported or contradictory statements, which are only made worse by the use of words such as "remarkable" (seemingly to also boost the importance of this work for a high IF journal). No need for that; this is a solid study without the need to use hyperbolas. I made many comments in the Discussion in which I am trying to challenge the authors to delve deeper in some portions of the wildlife literature and move away from grandiose but unsupported statements. In particular, the Discussion needs a much stronger presence of the actual results and how they relate to the assertions. Right now it reads largely disconnected from Results. 

- Lastly, the last section should contain less arm-waving and theory and more practical implications of this important work. What exactly do the authors propose as immediate steps to safeguard wildlife based on this analysis? Talking about SES and frameworks, and how everyone should work together is great, but not very realistic or useful in the near-term, which is what is needed to stop defaunation. I challenge the authors o think outside the box and find practical solutions to met this challenges based on their work.

In a nutshell, this is a great paper that needs some additional explanations and stepping back from big statements to more practical issues. Thank you for the opportunity to read your work.

---

## [Editor Report · Decision Letter 2]

27 May 2022

Dear Dr Bachmann,

Thank you for your patience while we considered your revised manuscript "Differences and commonalities in wildlife hunting across a Global South-North gradient" for publication as a Research Article at PLOS Biology. This revised version of your manuscript has been evaluated by the PLOS Biology editors and by the Academic Editor.

Based on our Academic Editor's assessment of your revision, we are likely to accept this manuscript for publication, provided you satisfactorily address the following remaining points and data and other policy-related requests.

IMPORTANT: Please address the following:

a) The Academic Editor strongly disagrees with your use of the word "global" to describe your study, and feels (like reviewer #2) that this may be misleading, especially in the title, and we ask that edit your manuscript to avoid this claim throughout.

b) Please change the title to remove the word "global" and to make it more explicit. We suggest "Analysis of the differences and commonalities in wildlife hunting across the Africa-Europe South-North gradient" or "Analysis of the differences and commonalities in wildlife hunting across the Africa-Europe South-North gradient reveals a strong effect of the human development index on sustainability" (if this claim is sufficiently strongly supported).

c) Your ethics statement currently says “The study was approved by the Ethics Council of the Max Planck Society, and we obtained informed consent from all participants.” If you have an approval number, please can you include this?

d) Please address my Data Policy requests below; specifically, we need you to supply the numerical values underlying Figs 1ABCD, 2ABC, 3ABC, 4ABC, 5ABCDEFGHI. Please supply these values as a supplementary data file. I note that your raw data are confidential (for understandable reasons); however, we still need the numerical values that immediately underlie the Figs.

e) Please also cite the location of the data clearly in each Fig legend, e.g. “The data underlying this Figure can be found in S1 Data.”

We expect to receive your revised manuscript within two weeks. 

*Published Peer Review History*

*Press*

Sincerely,

Roli Roberts

Senior Editor,

rroberts@plos.org,

PLOS Biology

DATA POLICY:

Regardless of the method selected, please ensure that you provide the individual numerical values that underlie the summary data displayed in the following figure panels as they are essential for readers to assess your analysis and to reproduce it: Figs 1ABCD, 2ABC, 3ABC, 4ABC, 5ABCDEFGHI. NOTE: the numerical data provided should include all replicates AND the way in which the plotted mean and errors were derived (it should not present only the mean/average values).

DATA NOT SHOWN?

---

## [Editor Report · Decision Letter 3]

13 Jun 2022

Dear Mona,

Thank you for the submission of your revised Research Article "Analysis of differences and commonalities in wildlife hunting across the Africa-Europe South-North gradient" for publication in PLOS Biology. On behalf of my colleagues and the Academic Editor, Andy Dobson, I'm pleased to say that we can in principle accept your manuscript for publication, provided you address any remaining formatting and reporting issues. These will be detailed in an email you should receive within 2-3 business days from our colleagues in the journal operations team; no action is required from you until then. Please note that we will not be able to formally accept your manuscript and schedule it for publication until you have completed any requested changes.

IMPORTANT: You'll see that we have slightly changed the beginning of your title, as previously requested ("Analysis of differences...) - this was largely to emphasise the fact that this is a piece of original research rather than a review article.

Sincerely, 

Roli

Senior Editor

PLOS Biology

rroberts@plos.org